# Study Protocol of a Randomized Controlled Trial of Home Modification to Prevent Home Fall Injuries in Houses with Māori Occupants

**DOI:** 10.3390/mps3040071

**Published:** 2020-10-23

**Authors:** Michael Keall, Hope Tupara, Nevil Pierse, Marg Wilkie, Michael Baker, Philippa Howden-Chapman, Chris Cunningham

**Affiliations:** 1He Kāinga Oranga, University of Otago, Wellington 6242, New Zealand; Nevil.Pierse@otago.ac.nz (N.P.); Michael.Baker@otago.ac.nz (M.B.); Philippa.Howden-Chapman@otago.ac.nz (P.H.-C.); 2Research Centre for Hauora & Health, Massey University, Wellington 6021, New Zealand; hope.tupara@gmail.com (H.T.); m.wilkie@massey.ac.nz (M.W.); chr1sc@me.com (C.C.)

**Keywords:** home modification, interventions, fall injuries, randomized controlled trial, indigenous health

## Abstract

Worldwide, injuries due to falls in the home impose a substantial burden and merit considerable effort to find effective prevention measures. The current study is one of very few randomized controlled trials that assess the effectiveness of home modification for preventing falls. It is the first carried out with a minority or indigenous community and focused on reducing inequities. Just over 250 households in Aotearoa, New Zealand, with Māori occupants were recruited in two strata, 150 from the Wellington region and 100 from the Taranaki region. These were randomly allocated to equally sized treatment and control groups within the respective regions, the treatment group receiving a package of home modifications designed to prevent falls at the start of the study, and the control group receiving the package at the end of the study. Injury data came from the Accident Compensation Corporation, a state-owned no-fault injury insurer. This provided coverage of virtually all unintentional injuries requiring medical treatment. Matched injury claims were made available for analysis once all identifying fields had been removed. These data will be pooled with data for Māori households from the already-conducted Home Injury Prevention Intervention (HIPI) study, which tested an identical intervention on the general population. In the analysis, the primary outcome measure will be fall injury rates over time, comparing treatment and control households, adjusting for the stratum and prior falls in the household. A secondary measure will be the rates of specific injuries, which are most likely to be prevented by the package of modifications tested. We anticipate that the findings will provide robust evidence for effective injury prevention measures that can reduce an important contributor to health inequities for indigenous populations such as the Māori.

## 1. Introduction

Globally, 1.4% of disability-adjusted life years (DALYs) due to disease and injury in 2010 were from fall injuries, which were the third most important injury type following road injury and self-harm [1]. In New Zealand, falls accounted for 2.9% of DALYs, which was only slightly less than road injuries (3.1%) [2]. Between 2014 and 2016, there was an average of 544 unintentional deaths due to falls per year [3]. Between 2014 and 2018, in a population of 5 million there were more than 24,000 hospitalizations annually for unintentional fall injuries [3] and according to data provided by the Accident Compensation Corporation (ACC), between 2011 and 2018, there were an average of 658,000 falls annually requiring some form of medical treatment, of which more than half were at home (where the setting was defined).

Given the importance of the home as a setting for falls, the home environment is a potential focus of prevention efforts to install safety features or to remove hazards. However, randomized controlled trials focused on the safety benefits of home modification are rare. A meta-analysis of such trials found that home safety assessments, with subsequent modification, were effective in reducing the rate of falls among older people living in the community (rate ratio 0.81, 95% confidence interval (CI) 0.68 to 0.97; 6 trials; 4208 participants) [4]. Our previous study, the Home Injury Prevention Intervention (HIPI) randomized controlled trial, found a reduction in home fall injury rates of 26% (95% CI 6–42%) [5], although the study was not powered to look at specific groups, such as the Māori, the indigenous people of Aotearoa, New Zealand. As Māori suffer from poorer health on average than other New Zealanders, measures that may potentially reduce health inequalities, such as home modification to reduce fall risk, need to be identified. The aim of the current study was to assess the safety benefits of home modification specifically for Māori. The trial was designed to generate evidence for intervention that could be rolled out as a national programme.

Outcomes studied when assessing fall prevention measures are often problematic as they require self-report of incidents, a measure that is subject to various recall-related issues that potentially affect the estimate of effectiveness [6]. A relatively complete record of medically-treated injuries avoids many such limitations. New Zealand has a national no-fault personal injury insurer, the Accident Compensation Corporation (ACC). ACC covers most of the treatment costs for injuries (regardless of fault) provided by doctors, dentists, physiotherapists, specialists, counsellors and other health professionals and 80% of lost earnings apart from the first week absent from work. Where the co-payment provided by ACC is insufficient to cover costs, the patient pays an additional surcharge. Although the ACC data provide a near-complete set of records of injuries requiring some form of medical or dental treatment, lower levels of access to medical treatment can lead to lower claim rates that underrepresent the underlying injury rates, as has been noted for Māori, for example [7]. Despite these minor limitations, ACC data provide a highly suitable outcome measure (counts of injuries receiving medical attention) to test the hypothesis that occupants of homes receiving the package of modifications will have a lower rate of home fall injuries over a subsequent two-year period.

## 2. Materials and Methods

### 2.1. Methods and Design

#### 2.1.1. Study Design

The Māori Home Injury Prevention Intervention (MHIPI) is a single-outcome randomized trial. The injury data to be analyzed are anonymized and the household is the unit of analysis. The household consists of all people who nominate the included household address as their home address. A secondary analysis will look at the odds of home fall injuries compared to other home injuries, which will account for age, which was found to be important in the analysis of the previously conducted HIPI study [5].

Ethical approval has been provided by the Massey University Human Ethics Committee (reference 4000022369).

#### 2.1.2. Target Population and Sampling

Participants were recruited by community providers from households in the Wellington and Taranaki Regions of New Zealand, some of whom recently received government-subsidized home insulation retrofitted to their homes, and others recruited from networks provided by iwi (Māori tribal groups), direct advertising in community newspapers, presentations to community groups and events, as well as personal contacts. To qualify for the study, houses needed to have at least one occupant who identified as Māori. Only those people who stated they intend to live at that address for the subsequent three years were eligible for participation, as the study seeks to evaluate the safety benefits of home improvements over that period. Only owner-occupiers were approached, as people renting houses tend to be a very mobile population in New Zealand, which does not suit the aims of the study. There were no other qualifying criteria (such as age, sex, health, previous falls, etc.). Two community trusts that mainly undertake insulation retrofitting recruited participants: WISE–Better Homes in Taranaki and the Sustainability Trust in Wellington. Following consent to be part of the study, which was obtained by the community trusts, 254 households were randomized to treatment and control groups by an Otago University statistician (see Figure 1).

## 3. Procedure

### 3.1. Home Modifications

A qualified builder assessed each house in the intervention group using a checklist to identify any hazards in the home that were within the scope of the treatment provided. The modifications done in the intervention houses over a 10-month period between December 2013 and October 2014, consisted of (where necessary): handrails for steps and stairs; some other minor repairs to steps; grab rails for bathrooms and toilets; provision of outside lighting; high-visibility and slip-resistant edging for outside steps; fixing of lifted edges of carpets and mats; non-slip bathmats; and slip resistant surfacing for outside surfaces such as decks. Where needed, smoke alarms were also installed, but are not relevant to the outcome measure. The same process was used for households in the control group once the study was completed, from late 2016 to early 2017. The average cost per house modified was approximately $NZ500. A 10% sample of households was contacted following the repairs to ensure that the treatment had been carried out as expected. All these households reported that the interventions were undertaken as specified in our records.

### 3.2. Outcomes

The primary outcome is rates of medically-treated unintentional falls per household per year. Secondary outcomes are medically-treated specific injuries (injuries that are potentially prevented by the treatment applied—see further details below) and costs of both these types of injuries, due to falls generally or due to the specific injuries. Costs are those borne by the health insurer, the ACC, as described above. Injury costs in relation to intervention costs will be analyzed to inform cost-benefit analyses of the intervention.

Rates will be estimated, firstly, for the MHIPI participants by themselves and, secondly, for the MHIPI participants plus all the Māori household members of the HIPI study, which shared an almost identical intervention. One important difference was that participants in the HIPI study consented to have the study team access their ACC injury records, which were assigned to individuals. To be consistent with the MHIPI data analysis, data for the Māori household members of the HIPI study will be aggregated to the household level.

A sensitivity analysis will be conducted of the odds of falls and the odds of specific injuries compared to all other medically treated home injuries. This will enable models to be fitted that control for age group, which was found to be an important factor in the analytical models used in the analysis of the HIPI study [5]. Such an analysis will lose a proportion of information compared to the primary analysis of injury rates because households without any home injury claims over the assessment period do not contribute data. As the current intervention being tested was designed to prevent injuries arising from falls, it can be assumed that other types of injuries that occur in the home (burns, scalds, poisoning, cuts, stings, strains, collisions, etc.) would be relatively unaffected by the intervention. Using an odds ratio as an outcome measure has the additional benefit of controlling for propensity to seek—or to be differentially referred for—medical treatment following an injury, which is known to vary between different population groups, and to be lower for Māori than for others [7]. As randomization was carried out at the household level and stratified only at the regional level, there is always the potential for the intervention and control groups to differ at random, according to important characteristics associated with fall rates, such as age. Analysis of the odds of a particular injury type or mechanism compared to other injury types or mechanisms, is common in road safety for the assessment of vehicle-related safety characteristics, for example [9].

In order to detect a change of 26% in the percentage of households with medically-treated injuries arising from falls (α of 0.05, 80% power), we calculated that a sample size of about 125 households in each of the two arms of the study (treatment and control groups) would be required.

#### 3.2.1. Defining Falls and Specific Injuries

In the context of self-reports of falls, Lamb et al. [6] recommended that a fall should be defined as ‘‘an unexpected event in which the participants come to rest on the ground, floor, or lower level’’ and participants in studies using self-reported falls were recommended to be asked “In the past month, have you had any fall including a slip or trip in which you lost your balance and landed on the floor or ground or lower level?’’ These guidelines were used to classify injuries from the ACC claims data.

Key information on fall injuries is provided by a free text field on the ACC claim form to describe what happened, generally a single sentence. Using these text descriptions in conjunction with other fields in the ACC claims data, coders will classify injuries as either falls or injuries specific to the package of modifications [10]. As well as referring to the text descriptions, coders consider the fields listing the activity immediately preceding the injury (e.g., running/walking), the “cause” (e.g., tripping or stumbling), any contact with objects/people/animals (e.g., ground/floor), any “external agency” (e.g., stairs/steps) and a general injury diagnosis (e.g., contusion). These fields help identify the specific injuries, which are those potentially preventable by the package of modifications applied, including virtually all injuries occurring on outside steps and most falls occurring in bathrooms.

Generally, the specific injuries are a subset of the fall injuries. Some injuries classed as specific occur from loss of balance not necessarily resulting in the subject being on the ground or floor. Such injuries occur when balance is lost and the subject collides with an object, causing injury, or some musculoskeletal injury occurs, such as muscle/tendon/ligament strain, as a consequence of loss of balance. The specific injuries generally exclude those arising from gardening, trips on furniture or children’s toys. Consistent with loss of balance, as an essential aspect of the Lamb et al. [6] falls definition, we exclude injuries occurring when the subject was jumping (such as trampoline injuries), undertaking sporting activities, or deliberate injuries (e.g., being pushed).

Falls in garages, falls from ladders and vehicle injuries (falls from bikes or cars) are excluded as these are not normally classified as injuries occurring in the home setting. We also exclude syncopal falls where the subject fainted and was consequently injured, as these do not involve loss of balance in the normal sense.

As for the HIPI study, a count will be made of the number of fall injuries and specific injuries occurring during the year prior to the intervention.

#### 3.2.2. Home Injury Claims Data for Participating Addresses

Injury counts will be obtained from ACC claims data for all home injuries, excluding deliberate injuries (assaults or self-harm), matched by the address fields of participants that we provide to ACC. Any form of matching has the potential for errors. Addresses in New Zealand are commonly specified in non-standardized forms that are difficult to match precisely. ACC have an identification number that is assigned to the claimant’s address, however it is not unusual for a given address to have more than one identification number, mainly arising from different ways that suburbs and towns are specified. This in turn is related to the number of different people in the household (each of whom may specify their address slightly differently to a given health provider). For the list of participants’ addresses we provided to ACC, they in turn furnished information on the number of ACC address IDs matched to the provided addresses, which was sometimes zero. One participating household’s address was matched to 24 ACC address IDs, which was clearly an error. This household will be excluded from any analyses (see Figure 1). There were a further eight households linked to two ACC address IDs, four linked to three IDs and one linked to five IDs. There is no feasibly mechanism by which the occurrence of multiple matches could be associated with the random assignment to intervention or control groups, which occurred after participants were recruited and visited. Multiple-matching of an intervention address could be expected to attenuate any intervention effect, as no change in injury risk can be expected for unmodified houses mistakenly classified as being in the intervention group.

Another form of matching error will arise where no match is made between a participant’s address and their ACC claims for home injury, because of errors in the specification of the addresses in either database. There were 35 participants’ addresses out of a total of 252 (14%) with no matching ACC claims, which provides an upper limit of the degree of such non-matching that occurred. Of these, 13 were in the intervention group. Some of these will not be in the ACC list of addresses because no occupant has had an injury resulting in an ACC claim while at that address. In the analyses, these households will be assumed to have zero injuries, including falls and specific injuries.

### 3.3. Analysis

#### 3.3.1. Period over Which Outcomes Are Studied

Injuries in the treatment group will be studied from the date of the intervention to the median date that modifications were carried out for the controls in the given region; injuries in the control group will be studied from the median date of interventions undertaken in the treatment group in the given region until the date modifications were carried out for that household. In situations where households did not have any modifications carried out because the household could not be contacted, median dates will be used as the specified modification date. For houses for which no modifications were necessary, the date that the house was visited and assessed will be used as the modification date. A small number of houses in the intervention group were modified, but details on the dates and the precise modifications applied have been lost. For these houses, median intervention dates will be used. All analyses will be conducted according to the intention-to-treat principle.

#### 3.3.2. Estimating Injury Rates

In the HIPI study [5], participants consented to researchers analyzing their ACC injury records as a condition of being part of the study, and these were provided by ACC for the analysis. For the MHIPI study, ACC provided anonymized injury records matched to addresses we supplied to them and all household members were potentially part of the study. The ACC injury data included the age group and sex of the claimant as well as an address identifier that was matched to information on the treatment arm and details of the modification applied once address information had been removed. Therefore, we cannot estimate fall injury rates per person-time as we have no adequate denominators. Instead, we will estimate injury rates at the household level. Although the data were anonymized for the MHIPI study, meaning that the total number of people whose injuries were potentially being assessed from the anonymized address lists was unknown, the mean number of consenting participants per household from the HIPI study was 2.2 overall, 3.1 for Māori and 2.1 for non-Māori households.

The treatment effect will be estimated by the SAS [11] procedure GENMOD by fitting a Poisson model with a log link function to injury counts at the household level with an offset that is the log of the length of the time interval over which the injury count was obtained. As with the HIPI study, where prior falls were an important predictor of falls during the trial period, a covariate in the model would be a count of fall injuries in the 365 days prior to the start of the trial for the given household. As it is possible that different tradesmen carrying out the intervention may undertake the modifications in slightly different ways, with differing degrees of effectiveness, a stratum variable will also be included as a covariate that discriminates between participants in Taranaki and those in Wellington. For the main analysis drawing on additional data for Māori households from the HIPI study, the stratum variable will take a third level. As the unit of randomization, the household, is also the unit for which rates are estimated, there is no clustering to be accounted for in this analysis.

As with the HIPI study analysis, we will also analyze rates of specific injuries (specific to the intervention) for the treatment and control groups, expecting the safety effect to be stronger for these injuries than for fall injuries, as it was for the HIPI study.

#### 3.3.3. Odds of Fall Injuries

In the case of the sensitivity analysis of the odds of a home fall, the treatment effect will be estimated by the ratio of the odds of home fall injuries vs. other home injuries for the treatment group compared to the controls. We will use SAS regression procedures to calculate crude and adjusted odds ratios accounting for clustering at the individual and household level, such as GENMOD [11] with a REPEATED statement, which fits a GEE model to adjust standard errors for the effects of clustering. As we are analyzing the odds of fall injuries, we would expected some factors leading to correlations of injury rates within households, such as propensity to seek medical treatment, to have less of an effect (as they affect both numerators and denominators equally), leading to a smaller intraclass correlation coefficient (ICC).

An odds ratio less than one will indicate a safety effect from the intervention. We will also analyze the odds of specific injuries vs. non-fall home injuries using the same model specification.

This trial is registered, number ACTRN12609000779279.

## 4. Discussion

This study is designed to explore a possible intervention that will inform decision markers looking for improvement in both health and inequalities. The research addresses an issue of inequity for Māori as Māori are more at risk from inadequate housing, because of low incomes, larger family sizes, and discrimination in private rental housing. Prior research has identified barriers for Māori in making ACC claims for injuries [7], leading to apparent lower injury rates, but potentially poorer longer-term outcomes because of lack of access to medical treatment for injuries. It would be expected that there is, therefore, a proportion of injuries currently not appearing in the ACC claims database, because of lack of access to medical treatment for such injuries. The intervention tested here can be expected to prevent a proportion of these untreated injuries, as well as the medically treated ones, which will generate further benefits, although they are difficult to quantify. This research is, therefore, vital for addressing a potential source of health inequities for Māori arising from injuries that are initially untreated.

This study, and the HIPI study that shared the same design, is limited in the way it can assess safety benefits separately for subsets of the intervention, i.e., particular types of modification carried out. This is because the study is designed to replicate the rollout of a potential injury prevention programme, which would take several effective interventions and apply them as a package. Such an approach is more efficient than doing each individual intervention separately, but it does limit our ability to assess individual aspects of the programme. Few specific types of modification were carried out in isolation, so inferences about safety effects of particular types of modifications are not feasible. For example, of the 436 homes modified in the HIPI study, on only one occasion was modification to the bathroom carried out without there being other modifications in the house [5]. Another limitation for assessing safety effects of specific modifications is that the treatment randomized was a package of modifications. Modifications within this package were not applied randomly, but were applied according to what was required. There are likely to be systematic differences between the behaviors, or individual factors, affecting fall risks of occupants of houses requiring a specific type of modification and houses not requiring that modification.

The current study is one of just a handful of randomized controlled trials on the safety benefits of home modification and the first study to our knowledge to focus on an indigenous population. We anticipate that the findings will provide evidence for effective injury prevention measures that can reduce an important aspect of health inequities.

## Figures and Tables

**Figure 1 mps-03-00071-f001:**
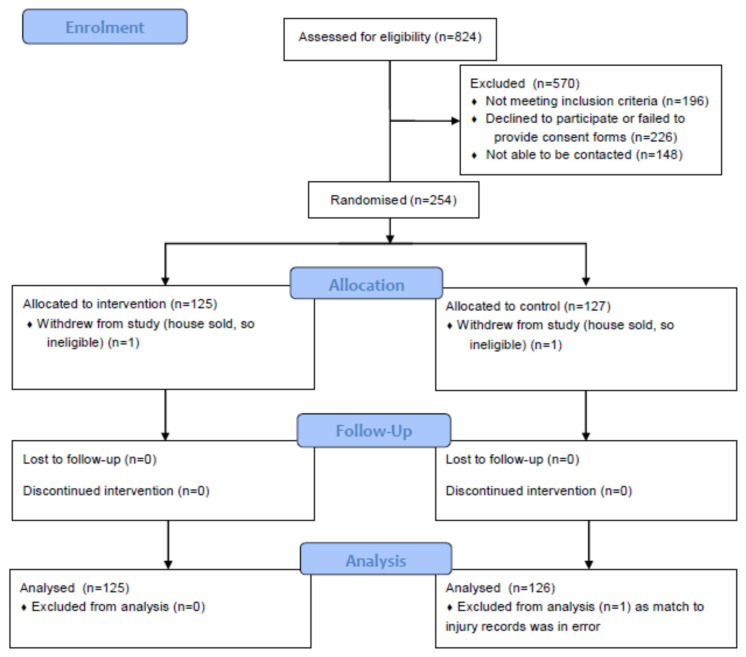
Flow diagram of participant numbers (households) according to CONSORT 2010 [8].

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
