# Peer review of "Study Protocol of a Randomized Controlled Trial of Home Modification to Prevent Home Fall Injuries in Houses with Māori Occupants"

_mps, 2020, doi:10.3390/mps3040071_

Round 1
Reviewer 1 Report
Generally, I have read your paper with great interest. It is a very sound paper, the idea is great and I would like to congratulate authors for the effort and approach used. My specific remarks and/or questions are stated below.
I think that the aim of the study shouldn’t been written as to provide sufficient statistical power for Maori. It sound strange, as any underpowered study has severe limitations. Your hypothesis is that Maori have poorer health (you have not explained why is it so) and you would like to see if home modification program for fall prevention works in this population. In that terms you have not been able to demonstrate clear connection between the last paragraph of the introduction and aims of the study. As a layman reader I don’t understand your intention with ACC data.
In study limitations you have stated “Only those people who stated they intend to live at that address for the subsequent three years were eligible for participation…” Where there any other exclusion/inclusion criteria for participation (e.g. age, sex, health condition, history of previous falls…).
Your intervention does not include any exercise related interventions such as balance and strength training etc. How or can you control for the confounding effect of such interventions on your own results?
Considering the fact that you will have a defined time variable what do you think about applying survival curves analysis where fall would constitute an event of interest and then do a log-rank test to compare CON vs.EXP group?
Could you elaborate little more about “A stratum variable will also be included as a covariate that discriminates between participants in Taranaki and those in Wellington.” Why is this needed? Where is the difference between two, and why controlling for this is important.
Author Response
Thank you for the constructive comments provided. The following has referees' comments in inverted comments and responses following. References to line numbers are for the tracked changes.
"I think that the aim of the study shouldn’t been written as to provide sufficient statistical power for Maori. It sound strange, as any underpowered study has severe limitations. "
Line 60-61 now reads: The aim of the current study was to assess the safety benefits of home modification specifically for Māori.
" In that terms you have not been able to demonstrate clear connection between the last paragraph of the introduction and aims of the study. As a layman reader I don’t understand your intention with ACC data."
To link the two sections more seamlessly, line 75-77 now reads: Despite these minor limitations, ACC data provide a highly suitable outcome measure (counts of injuries receiving medical attention) to test the hypothesis that occupants of homes receiving the package of modifications will have a lower rate of home fall injuries over a subsequent two-year period.
We have also changed the following paragraph so that it just states the study design (line 82): The Māori Home Injury Prevention Intervention (MHIPI) is a single-outcome randomized trial.
"In study limitations you have stated “Only those people who stated they intend to live at that address for the subsequent three years were eligible for participation…” Where there any other exclusion/inclusion criteria for participation (e.g. age, sex, health condition, history of previous falls…)."
In Line 103 we now state: There were no other qualifying criteria (such as age, sex, health, previous falls, etc.).
"Your intervention does not include any exercise related interventions such as balance and strength training etc. How or can you control for the confounding effect of such interventions on your own results?"
Although individuals will differ in terms of their propensity to fall (whether that arises from health conditions, exercise or other factors), the randomisation should mean that there will be a similar distribution of such propensities in each group (treatment and control).
"Considering the fact that you will have a defined time variable what do you think about applying survival curves analysis where fall would constitute an event of interest and then do a log-rank test to compare CON vs.EXP group?"
Because the data are anonymised, we cannot follow individuals over time. Fortunately, the analysis of fall rates (as we will do here) is a common method for assessing these sorts of environmental interventions.
"Could you elaborate little more about “A stratum variable will also be included as a covariate that discriminates between participants in Taranaki and those in Wellington.” Why is this needed? Where is the difference between two, and why controlling for this is important."
Line 249 now reads: As it is possible that different tradesmen carrying out the intervention may undertake the modifications in slightly different ways, with differing degrees of effectiveness, a stratum variable will also be included as a covariate that discriminates between participants in Taranaki and those in Wellington.
Reviewer 2 Report
It's ok for me.
the paper is fine written and the argument is interesting. The authors clearly explained the context and the experimental protocol. I didn't notice any grammatical or typing errors. I really don't have any other comments.
Author Response
Thank you for your comments
Reviewer 3 Report
This an excellent design. I only have several specific comments.
- It is unclear how the sample size is determined. Please specify the calculation details.
- It is also unclear how to ensure and quantify the complaince to the recommended interventions in the trial.
- Last, it may not be ture to exclude "falls from ladders and vehicle injuries". Probably, the authors should specify the range of "home injuries" (row 177). According to my own understanding, falls from ladders could also happen in the room.
Author Response
Response to comment 1.
We have added text in line 157-159: To to detect a change of 26% in the percentage of households with medically-treated injuries arising from falls (α of 0.05, 80% power), we calculated that a sample size of around 125 households in each of the two arms of the study (treatment and control groups) would be be required.
Response to comment 2
In lines 125-126, we have added the text in italics: A 10% sample of households was contacted following the repairs to ensure that the treatment had been carried out as expected. All these households reported that the interventions were done as specified in our records.
Response to comment 3
New Zealand home injury data recorded by ACC include a range of injuries not usually found in international studies for this setting. As vehicles are manoeuvred in areas that are within the boundaries of the home property, there are many vehicle injuries that are reported. In other countries, these would be considered traffic injuries. Likewise, injuries arising from home maintenance, sometimes involving ladders, are not commonly considered home injuries in other countries.